# The Scale of Intoxications with New Psychoactive Substances over the Period 2014–2020—Characteristics of the Trends and Impacts of the COVID-19 Pandemic on the Example of Łódź Province, Poland

**DOI:** 10.3390/ijerph19084427

**Published:** 2022-04-07

**Authors:** Anna Garus-Pakowska, Agnieszka Kolmaga, Ewelina Gaszyńska, Magdalena Ulrichs

**Affiliations:** 1Department of Nutrition and Epidemiology, Medical University of Lodz, 90-752 Lodz, Poland; agnieszka.kolmaga@umed.lodz.pl (A.K.); ewelina.gaszynska@umed.lodz.pl (E.G.); 2Department of Econometrics, Faculty of Economics and Sociology, University of Lodz, 90-255 Lodz, Poland; magdalena.ulrichs@uni.lodz.pl

**Keywords:** legal highs, new psychoactive substances, NPS, intoxication, COVID-19

## Abstract

Legal highs are new psychoactive substances (NPSs) which pose a high risk for human health, and the spread of the SARS-CoV-2 pandemic has changed peoples’ behaviours, including the demand for NPS. The aim of the study was to assess both the frequency of intoxication with NPS in Łódź province over the period 2014–2020, and the impact of the COVID-19 pandemic on developing this trend. An analysis was carried out of data on intoxications in Łódź province in the years 2014–2020 reported by hospitals. The medical interventions rate (MI) per 100,000 people in the population was calculated. The frequency of intoxications was compared taking sociodemographic variables into account, and the effect of seasonal influence on intoxications was calculated using the Holt–Winter multiplicative seasonal method. In the period considered, there were 7175 acute NPS poisonings in the Łódź province and 25,495 in Poland. The averaged MI rate between 2014–2020 was 9.45 for Poland and 38.53 for the Łódź province, and the lowest value was found during the COVID pandemic in the year 2020 (respectively, 2.1 vs. 16.94). NPS users were mainly young men of 19–24 years old from a big city. Most cases were registered at weekends and in summer months. The majority of intoxications were caused by unidentified psychoactive substances of legal highs (chi^2^ = 513.98, *p* < 0.05). The actual number of NPS-related poisonings in the Łódź province in 2020 was lower than the value extrapolated from trend analysis of data between 2014–2019. NPS use in Poland decreased during the pandemic. It should be noted that a decrease in the number of drug-related incidents can have more than one reason, e.g., preventive programs, increased awareness, or changes in the law. This paper advocates that, in addition to monitoring NPS-related intoxications, there is further investigation into the social, cultural, and behavioural determinants of NPS to facilitate targeted prevention programmes and the development of new medical treatments.

## 1. Introduction

“Designer drugs”, “smarts”, “boosters”, “part pills”, or “legal highs” are terms commonly used to refer to different substances of natural, semi-synthetic, and synthetic origin or their blends, which have alleged or true body–mind effects. Academic organisations maintain the term “New Psychoactive Substances” (NPS) to emphasise that they require current scientific research, in terms of the identification of these substances, their pharmacological effects, and their health and social impacts [1,2,3]. United Nations Office on Drugs and Crime (UNODC) defines new psychoactive substances as new drugs, substances of abuse that are not controlled, either in a pure or a preparation form, which may pose a serious public health threat [4].

These substances may have diverse contents, effects, metabolism, half-life, and intensity of side effects [4,5].

The unpredictability and diversity of NPS contents, and the lack of a list of ingredients on packaging, are frequent difficulties during diagnosis and treatment. The chemical content of legal highs is not available on most packaging, and on some packaging it is very often changed, or varies in different stores for the same product [6]. Without the knowledge of the exact chemical content of new drugs, doctors and toxicologists have severe problems in diagnosing patients with poisoning symptoms. In addition, legal highs are often taken together with non-prescription medicines, alcohol, or other psychoactive substances [2,7,8,9]. They are dangerous due to the huge lack of expertise on their toxicity and the narrow line between a safe dose and a lethal dose [4,10,11].

Following NPS use, a large variety of symptoms with different severities are seen, which depend on the dose and type of substance. The following symptoms have been described in the literature:-digestive system—diarrhoea, nausea, vomiting, and bleeding;-cardiovascular system—tachycardia, increased blood pressure, palpitation, and chest pain;-nervous system—headaches, dizziness, anxiety, irritability, convulsions, agitation, aggression, hallucinations, panic attacks, and coma;-metabolic disorders—hyperglycaemia, hyponatremia, hypokalaemia, and acidosis;-other—fever, diaphoresis, mydriasis, conjunctival congestion, hypothermia, and suicidal tendencies [9,12,13].

Nowadays, like drugs, NPSs can be found in tablet, capsule, powder, dried herbs, and chewable cube forms. Legal highs are administered most often orally, via smoking or inhalation [2,9]. According to Mazurkiewicz et al., smoking mixtures (75%) and tablets (12.5%) are the most popular forms of legal highs [14]. NPSs tend not to maintain their effect; however, long-term implications of their use are difficult to predict. People who use NPS in the initial phase develop euphoria, followed by behavioural disorders (selfishness, egocentrism, apathy), while the long-term impacts are still poorly understood. However, the available data show that long-term legal highs users may develop neurosis, psychosis manifested by anxiety, behavioural changes, and self-injurious behaviour [15,16,17].

The COVID-19 pandemic has changed peoples’ behaviours. The pandemic has also affected the addictive substance market from production to distribution by the introduction of different restrictions on the freedom of movement, work, leisure, and social meetings, which resulted in the modification of consumption patterns. According to the Global Drug Survey, a general feeling of stress caused by the pandemic (pandemic-related restrictions) increased the consumption of some drugs. On the other hand, the decrease in the consumption of psychoactive substances has been seen due to the restrictions in social life (fewer occasions to use drugs) [18].

The aim of the study was to assess the frequency of intoxication with in new psychoactive substances in Łódź province over the period 2014–2020.

The specific objectives of this study were as follows: (1) the comparison of data for Łódź province with those for Poland; (2) the assessment of the frequency of NPS intoxication submitted by hospitals in Łódź province in the years 2014–2020 including the sociodemographic characteristics of the affected; (3) the analysis of time periods with the largest number of NPS intoxications and the assessment of the influence of the COVID-19 pandemic on the spread of consumption/trying NPSs.

## 2. Materials and Methods

In Poland, it is the Chief Sanitary Inspectorate that keeps a register of legal high intoxications and suspected poisonings. The aim of the register is to inform the public about new psychoactive substances within the early warning system. The register is kept on the basis of information from hospitals which have provided care to patients with suspected intoxication. These are patients with acute poisoning, most often taken from the street by an ambulance. Patients with suspected poisoning are admitted to acute poisoning units or intensive care units. The clinical decision is made by a physician who completes the relevant documentation. Information about intoxication is passed on to the State Sanitary Inspector specific to the location of a given hospital. Then, it is sent in a collective form to the regional inspectors. The Regional Sanitary and Epidemiological Station send aggregated reports to the Chief Sanitary Inspectorate. The epidemiological surveillance system conducted in this way is a kind of a passive surveillance system. Hospitalized cases are entered in the register; therefore, the completeness of the data is full (according to hospital statistics). The data on poisoning were collected from the Regional Sanitary Inspector for Łódź province. The data were selected according to the annual highest rate of legal highs intoxication across the country.

The data on intoxications in Łódź province in the years 2014–2020 reported by hospitals were analysed.

A comparison was made between the regional and nationwide data (absolute figures and the rate of medical intervention). The medical interventions rate (MI) is the coefficient used by sanitary and epidemiological stations in Poland, which means the number of poisonings in which hospitalization was required per 100,000 people. In the calculation of the MI rate, the population was determined on the basis of the data from the Central Statistical Office, according to population figures as of 30 June each year. According to data from 30 June 2020, Łódź province was inhabited by about 2.4 million people.

The research involved 7175 people—886 women (12.35%) and 6289 men (87.65%)—who found themselves in hospitals in Łódź province because of NPS poisoning. To evaluate the frequency of intoxication, the sociodemographic variables were used, including sex, age, place of residence (small city—<100,000 residents; or a big city—>100,000 residents). A comparison of the number of legal highs intoxications was made depending on a day of the week, month, and year. In addition, it was taken into consideration whether the intoxications had been caused by psychoactive substances (ingredient of legal highs).

It was assumed that risky behaviours may have also changed due to the current epidemiological situation, the introduction of restrictions in public space, limitations on the number of people in meetings, isolation, and quarantine. It was noted that the lockdown caused a decrease in the number of intoxications. An attempt was made to estimate the level of decline resulting from the introduction of a general lockdown. It was assumed that a number of intoxications were observed in 2014–2019; on this basis, using the methods of time series analysis (Holt–Winters exponential smoothing) [19], an ex ante forecast of the number of intoxications was made for 2020, assuming that the current trends will not change.

The STATA/MP 16.1 (statistical software for data science) was used to analyse variables. The elements of the descriptive statistics were applied and the Pearson chi^2^ test was used to assess statistical significance of differences of the values of the examined variables. *p*-value < 0.05 was adopted as the level of significance.

In this report, only anonymous data were used; therefore, the research was not subject to obtaining consent of the relevant bioethics committee.

## 3. Results

During the analysed years 2014–2020 in Łódź province, 7175 medical interventions were recorded due to legal high intoxications, which accounted for 28.14% of all intoxications identified in that period in Poland (*N* = 25,495). The highest rate of intoxications was confirmed in 2015 (*n* = 1560 in Łódź province and *n* = 7281 in Poland), and the lowest rate of intoxications was confirmed in 2020 (*n* = 417 vs. *n* = 806). In turn, the highest rate of intoxications in Łódź province compared with the rest of Poland was registered in 2019 (37.8%), 2016 (34.01%), and at the beginning of the pandemic (2020), culminating at 51.75% of NPS intoxications reported in the Łódź province. The MI rate per 100,000 members of the population of the whole country over the period considered was 4 times lower, on average, than for Łódź province. The largest difference was noted in 2016 when the rate for Łódź province was 5.2 times higher that of Poland, and in 2020 when the rate was more than 8 times higher for Łódź province. Table 1 shows the detailed data.

Analysing the collected data in terms of sociodemographic characteristics of so-called legal highs victims it was stated that they were men in the vast majority of cases (*n* = 6289; 87.65%).

On the other hand, taking into account the age structure of the victims, the most deaths were registered for 19–24-year-old individuals (35.41%), and for 30–39-year-old individuals (24.63%). A 10-year-old child was the youngest patient and the oldest one was 76 years old. The majority of cases of intoxication were among 20-year-old individuals. They were men in the vast majority in all age groups (chi^2^ = 342.61, *p* < 0.05).

Most patients were the inhabitants of a large city (*n* = 4583; 63.87%). There were no differences between gender or a patient’s place of residence (chi^2^ = 4.44, *p* = 0.35). No significant differences were found between the number of intoxications for each day of the week (chi^2^ = 19.06, *p* = 0.087), even though at weekends there were 44.11% cases in total. The majority of cases were in July (*n* = 854, 11.9%), and there were more NPS intoxications in summer months than in winter (55% vs. 45%). The detailed results are shown in Table 2.

The vast majority of victims in particular years were men (chi^2^ = 44.91, *p* < 0.05). As in all analysed years, the majority of intoxications were due to unidentified psychoactive substances of legal highs (chi^2^ = 513.98, *p* < 0.05) (Table 3).

As we indicated, 5678 poisonings were caused by NPS of an unknown composition, which definitely influences the treatment success. Only 1497 poisonings were caused by substances with a known structural formula.

These were mainly synthetic cathinones (e.g., 3-MMC, pentedrone, alpha-PVP, 4-CMC) and synthetic cannabinoids (e.g., AB-CMINACA). Apart from those mentioned, two other groups of substances were identified: new benzodiazepines and synthetic opioids.

In addition, it should be emphasized that, among the poisonings, 225 cases of poisoning after consumption of various combinations of psychoactive substances were registered, which further determines the result of treatment. These were poisonings caused by mixtures of different NPSs or a combination of NPSs with a known drug/psychoactive substance. The numbers of substances detected in mixed poisonings are presented in Table 4.

Comparing the number of intoxications in March/April 2019 and 2020, one should notice a significant decrease in the number of patients affected by NPS (chi^2^ = 7.0325, *p* = 0.008) (Table 5). This decrease was observed at the same time as the introduction of the strict lockdown (March 2020).

The true number of poisoning cases in Łódź province in 2020 was lower than the estimated amount on the basis of the trend concerning the period before the pandemic. We can conclude that, on average, in 2020, there were approx. 26.59 cases fewer than it would appear based on the trend observed in 2014–2019. This corresponds to a reduction of about 91.5% in the total number of cases in 2020, see results in Figure 1.

## 4. Discussion

Each substance contained in legal highs has risks to human life, and the list of negative health effects is very long. There is no organ that would not be adversely affected [20]. The main factors influencing the harmfulness of legal highs are type of the mixture (e.g., NPS with other NPS, NPS with alcohol, or NPS with other psychoactive substance), uncertainty about the actual identity of the substance, and the amount of the toxic substance. Children and young people are the most vulnerable, at-risk population for the negative impacts of legal highs [7,21]. At the same time, the attractive and colourful packaging encourages youngsters to use legal highs [22]. In our research, children were not frequently hospitalized because of NPS intoxication; however, the results show that, sometimes, very young people use these substances (the youngest patient was 10 years old).

In 2019, 184 people died from severe NPS intoxications [23], which represents 0.7% of the total number of patients hospitalized due to acute poisoning in Poland [24]. The toxicity of designer drugs increases with the simultaneous consumption of alcohol or other psychoactive compounds, increasing the risk of death [25]. According to López-Pelayo et al., in 2017, one out of every six drug-related deaths in Europe resulted from NPS, and the number of deaths increased in relation to the number in 2016 [26]. In this respect, it is worth emphasising prevention, which should include consistent and interdisciplinary activities initiated by state institutions. Because of the young age of patients and the specific risk arising from the psychoactive substances, these activities should be carried out and participated in by parents and primary care physicians who are aware of the problem. Programmes targeted at young people on potential harmful effects of new psychoactive substances, making them aware of the danger and severe NPS-related consequences, should be of particular importance [9,14].

The new products, the consumption of which causes psychoactive effects, started to appear on the Polish market in 2007. The problem of NPS has been noticeable in Poland since the entry of a new product, Spice, in 2006. The people who were opening headshops perceived themselves as collectors, and the substances existed under the name of collection products, not for human consumption; that is why they were beyond the control of legislation. Shortly, a lot of stores appeared offering goods with substances not covered by the Act on Addictive Drugs, which is why the sale of these products was completely illegal. It was only later that legal action was carried out which resulted in the exclusion of the legal sale of psychoactive products on the Polish market. Production and marketing of legal highs have been prohibited in Poland since 27 November 2010, and this ban was extended to new psychoactive substances on 1 July 2015. In addition, the importation of these substances to the Republic of Poland has been banned by law [27].

The increase in intoxications in 2015 which we have observed in our research resulted from the change of the provisions of the law in 2015. The growing activity of dealers introducing so-called “legal highs” on the market in 2015 (supply growth and promotion) was caused by the wish to remove the stocks because of the planned amendment of the Act on Counteracting Drug Addiction, which placed 144 substances on the list of psychotropic substances and narcotic drugs. Information on identified psychoactive substances and the danger they pose is published on the Chief Sanitary Inspectorate website in Poland, and is communicated under the early warning system to the Information Centre for Drug Prevention. Then, these data are sent to the European Monitoring Centre for Drugs and Drug Addiction (EMCDDA) in Lisbon, which enables the exchange of information on dangerous substances entering other European countries. The prevalence of NPS use has been showing a downward trend since 2015. Many factors may have influenced this, including the changes in legal regulations and legislation increasing peoples’ awareness of the dangers of consuming NPSs. According to “Report on the state of drug addiction in Poland in 2020”, the percentage of respondents receiving offers to purchase, receive, or use these substances, as well as those actively evaluating their availability, is decreasing. Attention was also paid to the growing group of respondents assigning a high risk to even experimenting with NPS [28].

For several years, cathinone and synthetic cannabinoids have been identified in Poland as the most popular NPSs, comprising some 88% of NPSs identified in 2018 [29]. A similar situation was observed in other countries. A Hungarian study carried out by Csák et al. in 2017 among the rural male population (the average age was 25 years) showed that NPSs were easily accessible and bought, especially the synthetic cannabinoid receptor agonists (SCRAs): 79% admitted that SCRAs were easy to obtain. Both SCRAs and synthetic cathinones were used regularly. Up to 57% of SCRAs users and 37% of synthetic cathinones users had used the substances within the previous 30 days, at least once a week. The researchers observed that there is limited understanding among NPS users of consequences, social and health aspects, and available treatment options [30]. Korf et al. carried out a study in six European countries (Germany, Hungary, Ireland, the Netherlands, Poland, and Portugal), and they classified 3023 NPS users into three groups—socially marginalised, nightclub users, and social network users. Herbal blends and/or synthetic cannabinoids in Hungary, stimulants in the Netherlands, psychedelics in Portugal, and dissociative drugs and other NPSs in Germany were the most frequently used [31].

The study of Vincenti et al. describes the results obtained from the analysis of the hundreds of packages confiscated in Italy in 2020 which were thought to contain NPSs. Illegal psychoactive substances were identified in 92.6% of samples. Synthetic cathinone appeared to be the most common substance. According to the authors, the results are likely to reflect the general demand for stimulants, and many of them are used as substitutes for MDMA (3,4 methylenedioxymethamphetamine) [32]. Unfortunately, despite unceasing studies on the contents of NPSs, the majority of intoxications are caused by unknown psychoactive substances, as confirmed by our study (over 50% per year, 90.4% in 2017). The lack of identification of NPSs is a huge problem. Most of them were catalogued as unknown substances. The chemical composition of these compounds is constantly changing, and the market for psychoactive substances is developing rapidly. Within a few years, the NPSs offered by the market changes, and substances that were very popular in one year, disappear from the market in the next, and are replaced by new ones. For example, in 2019, a new cannabinoid called 5F-MDMB-PICA joined the group of the most popular NPSs [28]. Certainly, legal changes had a significant impact on the dynamics of this market. Despite continuous research, the phenomenon of NPS is still largely unknown, and it is necessary to improve analytical systems allowing for quick identification of NPSs. At the same time, in the legal provisions prohibiting the trading of NPS, it should be ensured that the definitions of compounds from the NPS groups include new variants of these substances.

NPS users are mainly young men (18–35 years old) using social networks, discussion forums, or blogs to share experiences and information on trustworthy online stores or taking prohibited substances at parties, nightclubs, and at electronic music festivals [3,31,33]. In our research, it was young men between 19 and 24 years old from large cities who were the most frequently hospitalized patients. Despite the lack of statistical significance, intoxications occurred more often at weekends and in the summer months, which may reflect the social character of taking stimulant drugs.

The outbreak of the COVID-19 pandemic, a severe respiratory disease, has fundamentally changed the lives of people worldwide, while making it difficult for illegal drug users to obtain and use drugs [34]. According to Di Trana et al., trading disruption of the identified chemical precursors caused by the pandemic may contribute to an increasing interest in homemade substances [35]. At the same time, it was shown that the pandemic dramatically influenced peoples’ mental health, which may enhance the growing interest in NPSs [36]. Similar conclusions were made by van Laar et al., indicating, from a survey, an increase in cannabis consumption in the Netherlands, both in terms of quantity and frequency [37]. In a study by Scherbaum et al., approximately 81% of respondents reported no change in drug consumption due to the COVID-19 pandemic. However, the study was conducted among people addicted to drugs such as cannabis, heroin, and cocaine [38]. An addicted person will often do everything to obtain the drug they are addicted to. Our research concerned people using NPSs. We observed a significant decrease in the number of NPS intoxications due to the COVID-19 pandemic, perhaps because of the change of lifestyle and the limitation of social contacts, including events such as drug parties. It is possible that this “recreational” character of using NPSs led to the significant decrease in the number of intoxications in Łódź province and in Poland during the COVID-19 pandemic. According to Arillotta et al., the COVID-19 pandemic influenced the market of psychoactive substances in different ways, including the price and availability [39]. In our study, we noticed a decline in NPS consumption during the COVID-19 pandemic. However, many factors could have influenced it, such as the limited social meetings and fewer drug parties during lockdown, but also preventive programs, increased awareness, and changes in the law.

Our study has some limitations. The data for analysis came from hospital statistics. This system is a passive data collection system, and our database is complete only if the patient was admitted to a hospital. We do not have (based on our database) information on the number of people exposed to NPSs or on the number of people who use NPSs. Another type of research would have to be carried out here, such as a randomly selected survey. According to data from population studies, the percentage of people aged 15–64 who had contact with drugs over the last year in Poland amounted to 5.4% [40]. However, it must be remembered that the credibility of the survey data is flawed. In the future, the research may be carried out in two ways—conducting a survey on the prevalence of NPS use and comparing the results to poisoning data from the reports of the Sanitary Inspectorate, collected in a similar timeframe.

## 5. Conclusions

The pandemic has reduced NPS use in Poland. However, it should be noted that a decrease in number of drug-related incidents can have more than one cause, e.g., preventive programs, increased awareness, or changes in the law. Intoxications with unknown psychoactive substances were dominant. In our research, the majority of users seem to be young men from big cities using NPSs socially and recreationally (implied by the number of intoxications at the weekend and during summer months). This paper advocates that—in addition to monitoring NPS-related intoxications—the social, cultural, and behavioural determinants of NPS use are investigated to facilitate targeted prevention programmes and the development of new medical treatments.

## Figures and Tables

**Figure 1 ijerph-19-04427-f001:**
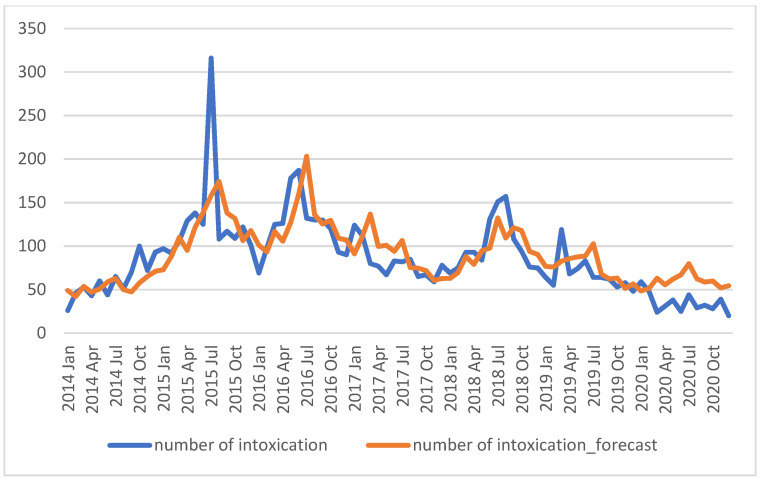
The number of intoxications and the number of intoxications forecasted over the years 2014–2020 using the Holt–Winters multiplicative seasonal method.

**Table 1 ijerph-19-04427-t001:** The number of legal high intoxications in Łódź province vs. Poland; the rate of medical interventions in Łódź province vs. Poland over the period 2014–2020.

	Number of Intoxications in the Following Years		MI/100,000	
Year	Łódź Province (% vs. Poland)	Poland	Łódź Province	Poland
2014	723 (29.83%)	2424	27.13	6.29
2015	1560 (21.43%)	7281	59.69	18.92
2016	1478 (34.01%)	4346	57.74	11.19
2017	979 (23.14%)	4230	38.54	11.00
2018	1206 (28.31%)	4260	44.28	11.08
2019	812 (37.80%)	2148	25.40	5.60
2020	417 (51.74%)	806	16.94	2.10
$\sum /\bar{x}$	∑ = 7175 (28.14%)	∑ = 25,495	$\bar{x}$ = 38.53	$\bar{x}$ = 9.45

∑—total; $\bar{x}$—average; MI—medical interventions rate.

**Table 2 ijerph-19-04427-t002:** The characteristics of patients in terms of their sociodemographic features and the characteristics of intoxications based on the analysed variables.

Variable		Number of Patients	Percent	Percent Cum
Sex				
	Female	886	12.35	12.35
	Male	6288	87.64	99.99
	No data	1	0.01	100.00
Age				
	<16	205	2.86	2.86
	16–18	718	10.01	12.87
	19–24	2541	35.41	48.28
	25–29	1538	21.44	69.72
	30–39	1767	24.63	94.35
	>39	351	4.89	99.24
	No data	55	0.77	100.00
Place of residence				
	City with population <100,000	2245	31.29	31.29
	City with population >100,000	4583	63.87	95.16
	No data	347	4.84	100.00
Day of the week				
	Monday	1102	15.36	15.36
	Tuesday	935	13.03	28.39
	Wednesday	973	13.56	41.95
	Thursday	1000	13.94	55.89
	Friday	971	13.53	69.42
	Saturday	1035	14.43	83.85
	Sunday	1159	16.15	100.00
Months				
	Summer (IV–IX)	3946	55.00	55.00
	Winter (X–III)	3229	45.00	100.00
NPS content				
	Unknown	5678	79.14	79.14
	Known	1497	20.86	100.00

**Table 3 ijerph-19-04427-t003:** The characteristics of intoxications with regard to a patient’s gender and NPS content in particular years.

	Year	2014	2015	2016	2017	2018	2019	2020		
Variable		*n* (%)	*n* (%)	*n* (%)	*n* (%)	*n* (%)	*n* (%)	*n* (%)	chi^2^	*p*
Sex										
	Female	113 (15.63)	187 (11.99)	120 (8.12)	142 (14.5)	155 (12.85)	112 (13.79)	57 (13.67)	44.91	<0.05
	Male	610 (84.37)	1373 (88.01)	1358 (91.88)	837 (85.5)	1051 (87.15)	700 (86.21)	360 (86.33)		
NPS content										
	Unknown	596 (82.43)	1285 (82.37)	1232 (83.36)	885 (90.4)	992 (82.26)	464 (57.14)	224 (53.72)	513.98	<0.05
	Known	127 (17.57)	275 (17.63)	246 (16.64)	94 (9.6)	214 (17.74)	348 (42.86)	193 (46.28)		
∑ (%) in the year		723 (100%)	1560 (100%)	1478 (100%)	979 (100%)	1206 (100%)	812 (100%)	417 (100%)		

chi^2^—Pearson chi^2^ test; *p*—statistical level of probability; ∑—total.

**Table 4 ijerph-19-04427-t004:** List of psychoactive substances in mixed poisonings (*n* = 225) *.

Psychoactive Substance	The Number of Cases of Mixed Poisonings in Which at Least One Psychoactive Substance Was Detected
THC (Tetrahydrocannabinol)	130
Amphetamine	125
Synthetic cathinones	42
Methamphetamine	23
Cannabimimetics	18
Cannabis	14
Mephedrone	11
Synthetic opioids	11
Ethyl alcohol	10
Synthetic hashish	7
Cocaine	5
LSD (lysergic acid diethylamide)	4
Methadone	4
Benzodiazepines	3
Heroin	2
Atropine	1
Dextromethorphan	1
Hallucinogenic mushrooms	1
Morphine	1

* the sum differs from the number of mixed poisonings (*n* = 225) because there were different patterns of taking psychoactive substances.

**Table 5 ijerph-19-04427-t005:** The comparison of the number of intoxications at the beginning of the lockdown in 2020 connected with the COVID-19 pandemic, respectively, over the same months.

Month	Year		Total		
	2019	2020		chi^2^	*p*
March	119	24	143	7.0325	0.008
April	68	31	99		
Total	187	55	242		

chi^2^—Pearson chi^2^ test; *p*—statistical level of probability.

## Data Availability

Not applicable.

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
