# Peer review of "The Scale of Intoxications with New Psychoactive Substances over the Period 2014–2020—Characteristics of the Trends and Impacts of the COVID-19 Pandemic on the Example of Łódź Province, Poland"

_ijerph, 2022, doi:10.3390/ijerph19084427_

Round 1

Reviewer 1 Report

The study by Garus-Pakowska  et al. describes a monitoring of the monitor the number of intoxications with NPS in the region Łódź between 2014-2020, and the impact of the Covid-19 pandemic on the number of NPS intoxications. The Medical Interventions rate (MI) adjusted for socio-demographic and seasonal variables was 9.45 per 100,000 for Poland and 37.1 for the region Łódź. Lowest rates of NPS intoxications were observed during the pandemic in 2020 (respectively 2.1 vs. 6.94). In most cases the NPS could not be identified (chi2=513.98, p<0.05). Based on their results, the authors advocate to investigate social, cultural and behavioral determinants of NPS to facilitate targeted prevention programmes and the development of new medical treatments.

The study has several points of weakness.

Major points

  1. The observations are few and not very solid. I have serious doubts about the quality and the implications the authors deduce from the data. Weakest point is that the identity of most, if not all NPS is not known. This is a severe omission knowing that NPS have a large variety in pharmacological profile.
  2. Introduction and discussion is not critical and contains (partly) wrong information.
  3. The conclusion does not match with the data presented.
  4. English is very poor and must be improved.

Minor points

Abstract:

  1. Legal highs are new psychoactive substances (NPS) which pose a high risk for human 12 health and life and the spread of Sars-Cov-2 pandemic has changed the contemporary world. Delete “and life”, because no fatal cases are reported here. Also delete “has changed the contemporary world”, because this is beyond scope and demagogic.
  2. What is “Voivodship”? Please use a common name. E.g. “the region Łódź”.
  3. “reported by medical entities” is too vague. Please describe properly (SHE stations ?). Similar: “Medical Interventions rate” is a vague term. In addition: Medical Interventions rate of WHAT ? Health incidents rate? Similar “the forecast number” might be rephrased as “effect of seasonal influence on”.
  4. “On average the MI rate was in the analysed period 9.45 for Poland and 37.1 for the analysed voivodship and it was the lowest over the pandemic period….” could be replaced by: “The averaged MI rate between 2014-2020 was 9.45 for Poland and 37.1 for the region Łódź with lowest value was found during the Covid pandemic in the year 2020….”
  5. “The statistical patient” is an awkward term.
  6. “The true number of poisoning cases in Łódź Voivodship in 2020 was lower than the estimated amount on the basis of the trend concerning the period before the pandemic.” This line is very confusing and poorly worded. Did you mean: “The actual number of NPS-related poisonings in the region Łódź in 2020 was lower than the value extrapolated from trend analysis of data between 2014-2019.”?
  7. The last line is incomprehensible. Please rephrase. Perhaps: “NPS use in Poland decreased during the pandemic. It is advocated to investigate, in addition to monitor NPS-related intoxications, social, cultural and behavioral determinants of NPS to facilitate targeted prevention programmes and the development of new medical treatments.”?
  8. Very poorly written. For instance: “the lack of the list of ingredients on the labelling”. I have never heard of labelling ingredients related to recreational drugs.
  9. Line 46-52 is also not correct. Some NPS have a narrow safety margin, whereas others don’t. Main problem in emergency is that the identity is not known (not the content of the pill, nor ill-labelling). Main problem for safety is that long-term effects of most NPS are not known.
  10. To list the most common complaints of NPS is nonsense, regarding NPS do not belong to one class of drugs. It would be the same to list the most common complaints of all drugs used for cardiovascular diseases !
  11. Line 81: “…the frequency of intoxication with substitutes….” What is substitutes? Variants of NPS?
  12. Line 93. “medical entities” please explain which entities (SEH, forensic doctors, academic toxicology services, hospital?).
  13. Would be relevant to know how many inhabitants the Łódź Voivodship has.
  14. The definition of “suspected substitute drug poisoning” is fully unclear. Are we talking now about NPS or what ? And how is “legal highs” defined? Is it identical with NPS? Similar: “psychoactive substance”. What is that ?
  15. Line 115. “anti-health behaviors” is an awkward term. Replace please.
  16. Line 117-119. “That is why, the calculation estimated the number of legal highs intoxications if the pandemic had not forced a change in social behavior.” is incomprehensible.
  17. Line 155. “The great number of patients are the inhabitants of a large city” > Most patients are the inhabitants of a large city.
  18. Between 2014-2015 the rate of intoxications more than doubled ! This should be addressed ! What was the reason? Secondly, a see a clear downward trend in intoxications between 2014-2020. What is the reason?
  19. Line 188. The main factor influencing the harmfulness of legal highs is uncertainty about long-term adverse effects !! Body weight is in this respect fully irrelevant. Also add: uncertainty about the actual identity. Mentioning ‘immunocompromised patients’ is fully irrelevant here as NPS do not act on immune function. By the way, patients who are severy ill should refrain from using recreational drugs anyway.
  20. Line 200-201. Comparing the mortalities among other drug, medicine and alcohol users, the majority of people who die are legal high addicts [25]. I cannot read this paper, but it would surprise me if this is correct. In addition, I think that NPS in general are not addictive, except for opioids. Please note that “number of deaths involving NPS” does not implicate a causal relation i.e., it is not per se the NPS that caused the lethality !
  21. A number of publications are available about changed drug use during Covid. These should be discussed. E.g. van Laar et al. Cannabis and COVID-19: Reasons for Concern. Front Psychiatry 2020 Dec 21;11:601653;  Molly Carlyle. Changes in Substance Use Among People Seeking Alcohol and Other Drug Treatment During the COVID-19 Pandemic: Evaluating Mental Health Outcomes and Resilience. Subst Abuse 2021 Dec 6;15:11782218211061746.
  22. The many double open spaces are disturbing.

Author Response

Rev.1.

We thank the reviewer for suggestions, and for advice on improving and developing our article.

Below we present our answers point by point. 

Major points

  1. The observations are few and not very solid. I have serious doubts about the quality and the implications the authors deduce from the data. Weakest point is that the identity of most, if not all NPS is not known. This is a severe omission knowing that NPS have a large variety in pharmacological profile.

Response: Thank you for your attention. We cannot agree with the too few observations, with as many as 7,175. On the other hand, we certainly agree with the reviewer that the identification of NPS is crucial to the implementation of appropriate treatment. Data was collected from official registries which indicated that during the period considered 1,497 poisonings were due to unknown NPS. And the whole problem with legal highs is that they contain an unknown psychoactive substance. Hence the strong emphasis on the identification of chemical compounds.

We have added the appropriate text to the results part:

Results:

As we indicated, 5,678 poisonings were caused by NPS of an unknown composition, which definitely influences the treatment success. Only 1,497 poisonings were caused by substances with a known structural formula.

These were mainly synthetic cathinones (e.g. 3-MMC, pentedrone, alpha-PVP, 4-CMC) and synthetic cannabinoids (e.g. AB-CMINACA). Apart from those mentioned, two other groups of substances were identified: new benzodiazepines and synthetic opioids.

In addition, it should be emphasized that among poisonings, 225 cases of poisoning after consumption of various combinations of psychoactive substances were registered, which further determines the result of treatment. These were poisonings caused by mixtures of different NPS or a combination of NPS with a known drug / psychoactive substance.

  1. Introduction and discussion is not critical and contains (partly) wrong information.

R: We improved the introduction, added the "Limitations" section to the discussion.

  1. The conclusion does not match with the data presented.

R: We improved the conclusions.

  1. English is very poor and must be improved.

R: We hope you find the text readable. We took into account the comments of the reviewer below. The text was linguistically checked, for which we also thanked in the manuscript.

Below are the answers to individual points: 

Abstract:

  1. Legal highs are new psychoactive substances (NPS) which pose a high risk for human health and life and the spread of Sars-Cov-2 pandemic has changed the contemporary world. Delete “and life”, because no fatal cases are reported here. Also delete “has changed the contemporary world”, because this is beyond scope and demagogic.

R: Thank you for your attention. We changed as suggested by the reviewer.

  1. What is “Voivodship”? Please use a common name. E.g. “the region Łódź”.

R: The word „voivodship” is an administrative unit in Poland, thus translated by the Great Oxford Dictionary; at the same time, IJERPH publishes articles in which the authors include the word "voivodship" in both the text and article titles (for example: https://doi.org/10.3390/ijerph16142601  ,  https://doi.org/10.3390/resources11020023, https://doi.org/10.3390/ijerph15112388, and many others. 

It is the area administered by a voivode (Governor) in several countries of central and eastern Europe. The word "province" has a similar meaning, but certainly not a „region”.

  1. “reported by medical entities” is too vague. Please describe properly (SHE stations ?). Similar: “Medical Interventions rate” is a vague term. In addition: Medical Interventions rate of WHAT ? Health incidents rate? Similar “the forecast number” might be rephrased as “effect of seasonal influence on”.

R: Thank you for your attention. We changed as suggested by the reviewer. “Medical Interventions rate”  is the coefficient used by sanitary and epidemiological stations in Poland = the number of poisonings in which hospitalization was required per 100,000. people. The term is explained in the text in the methodology section.

  1. “On average the MI rate was in the analysed period 9.45 for Poland and 37.1 for the analysed voivodship and it was the lowest over the pandemic period….” could be replaced by: “The averaged MI rate between 2014-2020 was 9.45 for Poland and 37.1 for the region Łódź with lowest value was found during the Covid pandemic in the year 2020….”

R: We changed as suggested by the reviewer. Thank you.

  1. “The statistical patient” is an awkward term.

R: We changed the phrase.

  1. “The true number of poisoning cases in Łódź Voivodship in 2020 was lower than the estimated amount on the basis of the trend concerning the period before the pandemic.” This line is very confusing and poorly worded. Did you mean: “The actual number of NPS-related poisonings in the region Łódź in 2020 was lower than the value extrapolated from trend analysis of data between 2014-2019.”?

R: We changed as suggested by the reviewer. Thank you.

  1. The last line is incomprehensible. Please rephrase. Perhaps: “NPS use in Poland decreased during the pandemic. It is advocated to investigate, in addition to monitor NPS-related intoxications, social, cultural and behavioral determinants of NPS to facilitate targeted prevention programmes and the development of new medical treatments.”?

R: We changed as suggested by the reviewer. Thank you.

  1. Very poorly written. For instance: “the lack of the list of ingredients on the labelling”. I have never heard of labelling ingredients related to recreational drugs.

R: We strongly agree with the reviewer. We in the text just emphasize this aspect of the lack of labeling, the lack of a list of ingredients.

  1. Line 46-52 is also not correct. Some NPS have a narrow safety margin, whereas others don’t. Main problem in emergency is that the identity is not known (not the content of the pill, nor ill-labelling). Main problem for safety is that long-term effects of most NPS are not known.

R: We agree with the reviewer. We do not write that the lack of knowledge of the composition is the main problem. We list a number of problems related to NPS. We also indicate „They are dangerous due to the huge lack of expertise on their toxicity, and the narrow line between safe and a lethal dose [4, 10-11]”.

  1. To list the most common complaints of NPS is nonsense, regarding NPS do not belong to one class of drugs. It would be the same to list the most common complaints of all drugs used for cardiovascular diseases !

R: We would like to kindly note that this is not an article about the clinical symptoms that occur with the consumption of certain chemicals. However, there are some common features, symptoms that make doctors suspect that the patient is under the influence (or may be under the influence of) drugs, and these symptoms are quoted based on the literature. We have added an explanation in the text.

  1. Line 81: “…the frequency of intoxication with substitutes….” What is substitutes? Variants of NPS?

R: In Poland, "substitute" is synonymous with a new psychoactive substance / legal high. Indeed, it may be incomprehensible to the reader, we automatically used a Polish synonym. Thank you for paying attention to this point. We corrected the word on NPS.

  1. Line 93. “medical entities” please explain which entities (SEH, forensic doctors, academic toxicology services, hospital?).

R: In both the executive summary and throughout the text, we have corrected the word entities to hospitlas. Thank you.

  1. Would be relevant to know how many inhabitants the Łódź Voivodship has.

R: We have added this information in the text. Thank you.

  1. The definition of “suspected substitute drug poisoning” is fully unclear. Are we talking now about NPS or what ? And how is “legal highs” defined? Is it identical with NPS? Similar: “psychoactive substance”. What is that ?

R: As mentioned above, we have changed the word substitute to NPS. This could be confusing to international readers. Thanks for your comment. The definitions of legal high and NPS are included at the beginning of the manuscript in the Introduction section.

  1. Line 115. “anti-health behaviors” is an awkward term. Replace please.

R: We have changed the word.

  1. Line 117-119. “That is why, the calculation estimated the number of legal highs intoxications if the pandemic had not forced a change in social behavior.” is incomprehensible.

R: In fact it's also written further on so we decided to delete that sentence. Thank you.

  1. Line 155. “The great number of patients are the inhabitants of a large city” > Most patients are the inhabitants of a large city.

R: We have corrected the sentence.

  1. Between 2014-2015 the rate of intoxications more than doubled ! This should be addressed ! What was the reason? Secondly, a see a clear downward trend in intoxications between 2014-2020. What is the reason?

R: Yes, this is a very important point. We wrote about it in the Discussion section. We have also added a text corresponding to the second point of the reviewer.

  1. Line 188. The main factor influencing the harmfulness of legal highs is uncertainty about long-term adverse effects !! Body weight is in this respect fully irrelevant. Also add: uncertainty about the actual identity. Mentioning ‘immunocompromised patients’ is fully irrelevant here as NPS do not act on immune function. By the way, patients who are severy ill should refrain from using recreational drugs anyway.

R: We changed as suggested by the reviewer. Thank you.

  1. Line 200-201. Comparing the mortalities among other drug, medicine and alcohol users, the majority of people who die are legal high addicts [25]. I cannot read this paper, but it would surprise me if this is correct. In addition, I think that NPS in general are not addictive, except for opioids. Please note that “number of deaths involving NPS” does not implicate a causal relation i.e., it is not per se the NPS that caused the lethality !

R: The reported mortality data are supported by literature. We changed the text as suggested by the reviewer to make it more readable.

However, when it comes to the risk of addiction, it is recognized that NPS, even if they are not physically addictive, carry a risk of psychological dependence. The publication describes it very well:” Legal highs: staying on top of the flood of novel psychoactive substances” by David Baumeister, Luis M Tojo, Derek K Tracy. DOI: 10.1177/2045125314559539.

There are many publications about deaths caused by the use of synthetic psychoactive substances, well cited in the following: Weinstein Aviv M., Rosca Paola, Fattore Liana, London Edythe D. „Synthetic Cathinone and Cannabinoid Designer Drugs Pose a Major Risk for Public Health”, Frontiers in Psychiatry, 2017, https://www.frontiersin.org/article/10.3389/fpsyt.2017.00156 DOI=10.3389/fpsyt.2017.00156 

  1. A number of publications are available about changed drug use during Covid. These should be discussed. E.g. van Laar et al. Cannabis and COVID-19: Reasons for Concern. Front Psychiatry 2020 Dec 21;11:601653;  Molly Carlyle. Changes in Substance Use Among People Seeking Alcohol and Other Drug Treatment During the COVID-19 Pandemic: Evaluating Mental Health Outcomes and Resilience. Subst Abuse 2021 Dec 6;15:11782218211061746.

R: We agree, of course, that in response to the ongoing pandemic, there are publications relating to its impact on people's behavior. We mentioned it and discussed some. We have added the proposed manuscript to the discussion section.

  1. The many double open spaces are disturbing.

R: It has been corrected.

Reviewer 2 Report

The manuscript of Garus-Pakowska et all. reports an epidemiological study on alleged NPS intoxication in Poland. However, there is a lack of clear data analysis. In particular, the NPS held responsible for poisoning are not reported. It is not reported whether it is intoxication with single substance or poly-drug intake of different substances. Presence of ethyl alcohol ?. This information is necessary to be able to understand if it is intoxication from NPS or other psychoactive molecules (amphetamines, opioids, etc ...). The type of intoxication is missing ...

Author Response

Dear Reviewer,

Thank you for your valuable suggestion. In our manuscript, we emphasized that most poisonings were caused by unknown chemicals in the afterburner. Generally, the main problem with taking NPS concerns the lack of knowledge of the content because it leaves the doctors of the wards that receive patients with acute poisoning with their hands tied. And this is what we wanted to emphasize.

However, it will indeed be interesting to add information on the most common active substances isolated - of course for those cases where the substance is known. Some poisonings were caused by taking the afterburner with a known composition, some poisoning - by taking a mixture of different NPS, and some were caused by taking the afterburner together with a known drug. It is difficult to systematize these results, because in principle how many patients - so many combinations of psychoactive substances.

In the case of legal highs, naming is quite important, but also complicated. On the one hand, we are dealing with the names of substances, on the other - products containing psychoactive substances. Yet another codification uses the initials of the scientists who first synthesized these substances, the abbreviations of the universities or pharmaceutical companies where they were synthesized, and many more.

Therefore - agreeing with the reviewer - we provided the main substances identified in the NPS in our study and presented the results concerning mixed poisonings in which at least one psychoactive substance was known.

We have added the appropriate text to the results part: - please find in the attachment.

Round 2

Reviewer 1 Report

Thank you for the answers. The paper remains disappointing, despite the improvements.

  1. “……the spread of Sars-Cov-2 pandemic has changed the contemporary world people 13 behaviors, including the demand on NPS…” No data have been presented endorsing this statement. Please delete from the abstract. Voivodship is not an English word, please use the synonym “province” as used in Methods and Results section. Also in “Voivodship Sanitary Inspector for Łódź Voivodship.”, line 114-115. “Łódzkie” is incorrect English grammar, please correct.
  2. “NPS use in Poland decreased during the pandemic.” No evidence is presented in the results endorsing this statement. Please delete from abstract. Note that a decrease in number of drug-related incidents can have more than one reasons (e.g., increased awareness, better purity, less drug parties during lock-down, in addition to lower prevalence of use). This point indicates the authors are not critically interpretating their results. See also (e.g.) lines 211-214
  3. Very poorly written. For instance: “the lack of the list of ingredients on the labelling”. I have never heard of labelling ingredients related to recreational drugs. The authors must address all points raised by the reviewers. If neglected, this is waist of my valuable time.
  4. “The symptoms of legal highs intoxication may resemble those of typical drugs.” This line is completely meaningless and must be deleted.
  5. Similarly, “NPS cause characteristic symptoms common to the entire group of compounds, but they may vary in severity depending on the dose and type of substance.” Also meaningless and not informative. Suggestion: “Following NPS use, a large variety of symptoms with different severity are seen which depend on the dose and type of substance”.
  6. “Young legal highs users are less educated, they are often unemployed and less satisfied with life [18].” Not only is this statement wrong, but the reference is certainly not describing this information. Moreover cannabis is not an NPS ! Typical for young NPS-users is, in my view, their relative high curiosity, high risky behavior and thrill-seeking behavior.
  7. Line 122. “the number of poisonings in which hospitalization was required”. Were all cases indeed hospitalized or did all cases attend to Emergency Departments (’First Aid’) of the hospital and partly send home ?
  8. “According to Global Drug Survey a general feeling of stress caused by the pandemic (pan-88 demic-related restrictions) increased the consumption of hemp products and benzodiazepine. On the other hand, the decrease in demand on addictive has been seen due to the restrictions in social life [19].” Again, cannabis and Benzo’s are no NPS. Please delete from MS. What is “demand on addictive”?
  9. Table 1: please use 1 decimal, not two.
  10. Line 197. “new benzodiazepines”. This is a rather old class of drugs. What is new about them? Probably, you mean benzodiazepines newly appearing on the recreative drug market?
  11. Line 197. synthetic opioids. It would be of interest to know if this is only (mainly) fentanyl ? Or also oxycodone?
  12. Table 4. Thanks for inserting this crucial information. As the main topic is NPS (cf. Title), please also indicate in the table NPS (Y/N). E.g., alcohol is no NPS. Question: does Table 4 depict only substances detected in mixed poisonings? (cf. line 202).
  13. Lines 211-214. Comparing …..(Table 5). NO ! No evidence is presented for this statement. There may be an association in time, but no causality was shown !! Please rephrase.
  14. Line 230-231. “type of the mixture”. Are all NPS mixtures ??
  15. Line 236-238: Redundant lines, because you apparently don’t dispose of prevalence of use data of NPS among Polish adolescents. Recommend to delete these lines. (Or suggest: “Despite that very young people have been reported to frequently use NPS, only few children in our sample were hospitalized.)
  16. Line 241. Suggest to replace ‘new drugs’ here by ‘NPS’. ‘New drugs’ were not defined before.
  17. Moreover, line 239-241. The number of acute poisoning in Poland is not of any interest here. Suggest: “In 2019, 184 people died from severe NPS intoxications [25] which represents some 0.7% of total number of hospitalized patients due to acute poisoning in Poland.”
  18. Line 264. “both substitutes and new psychoactive substances”. What is meant by substitutes ?
  19. 284-286. Suggest: “For several years, cathinone and synthetic cannabinoids have been identified in Poland as the most popular NPS, comprising some 88% of NPS identified in 2018 [30].”
  20. Line 290 use plural forms: Both SCRAs and synthetic cathinones
  21. Line 304-305. “many of them are used as substitutes MDMA (3,4 methylenedioxy-methamphetamine) [33].” Please rephrase (substitutes of MDMA)??
  22. Line 316-318. Fully redundant line, and not endorsed by data. Please remove.
  23. Line 348, The pandemic has reduced reduces NPS using in Poland. NO ! see point 13.
  24. AGAIN: Remove all double open spaces. E.g. line 312.

Author Response

Dear Reviewer,

thank you for your re-opinion; in the attachment we present our answers point by point.

Reviewer 2 Report

The work has been implemented.
However, my peplexity remains on the fact that the intoxications are not supported by an instrumental analysis.
Most of them are cataloged as unknown substances. The authors therefore state that the consumption of NPS is decreasing. Most likely various unknown substances are NPS that have not been analytically identified ...
This aspect is of great interest to highlight how the NPS phenomenon is still little known and an enhancement of analytical systems is essential.
Authors should discuss this aspect in their discussion.

Author Response

Dear Reviewer,

Thank you for your interesting attention. We wrote our opinion in the Discussion section. See lines 321-332.